# Inferring flavor mixtures in multijet events

Ezequiel Alvarez[*1] and Yuling Yao[†2]

[1]*ICAS, ICIFI-Conicet UNSAM, 25 de Mayo y Francia, San Martín (1650), Buenos Aires, Argentina*
[2]*Flatiron Institute, New York, NY 10010, USA*

## Abstract

Multijet events with heavy-flavors are of central importance at the LHC since many relevant processes—such as $t\bar{t}$, $hh$, $t\bar{t}h$ and others—have a preferred branching ratio for this final state. Current techniques for tackling these processes use hard-assignment selections through $b$-tagging working points, and suffer from systematic uncertainties because of the difficulties in Monte Carlo simulations. We develop a flexible Bayesian mixture model approach to simultaneously infer $b$-tagging score distributions and the flavor mixture composition in the dataset. We model multidimensional jet events, and to enhance estimation efficiency, we design structured priors that leverages the continuity and unimodality of the $b$-tagging score distributions. Remarkably, our method eliminates the need for a parametric assumption and is robust against model misspecification—It works for arbitrarily flexible continuous curves and is better if they are unimodal. We have run a toy inferential process with signal *bbbb* and backgrounds *bbcc* and *cccc*, and we find that with a few hundred events we can recover the true mixture fractions of the signal and backgrounds, as well as the true $b$-tagging score distribution curves, despite their arbitrariness and nonparametric shapes. We discuss prospects for taking these findings into a realistic scenario in a physics analysis. The presented results could be a starting point for a different and novel kind of analysis in multijet events, with a scope competitive with current state-of-the-art analyses. We also discuss the possibility of using these results in general cases of signals and backgrounds with approximately known continuous distributions and/or expected unimodality.

---

[*]sequi@unsam.edu.ar
[†]yyao@flatironinstitute.org

# 1 Introduction

The Large Hadron Collider (LHC) is one of the greatest machines ever built by mankind. Every knowledge, every available technology, and every to-be-available technology have been fully coordinated and exploited to give rise to a Machine that gives the maximum of it in all of its aspects. This has been confirmed by its outstanding performance along the years, including the discovery of the Higgs boson [1, 2]. Having more than a decade ahead, with a few Standard Model milestones and the chimera of unforeseen discoveries in the horizon, the physics side of the LHC is challenged to exploit to the fullest every bit of data and information available with tools and techniques at the state-of-the-art level. This work is directly aimed at contributing in this direction by making the most of the mutual information contained at the event-by-event level, exploiting prior information in different ways, and also exploring how to reduce systematic uncertainties by novel data-driven methods. The ideas deployed in this article are along the lines of considering maximizing the potential of LHC data as a new frontier in High-Energy Physics, the information frontier.

To this end, we focus on multijet events and the challenge of finding out their composition in terms of classes of possible flavor combinations. This kind of problem occurs often at the LHC and it has the added complication that simulations often have problems reproducing this type of events [3, 4]. We focus on heavy-flavor tagging, which is of main interest at the LHC because of its connection to heavy particles such as the top-quark and the Higgs boson. This kind of problem is relevant for final states such as for instance $t\bar{t}$, $hh$, $tW$ and $Zh$ among others, since the signal and its corresponding backgrounds in the multijet channel end up defining a multijet selection of events with bottom-quark jets, or $b$-jets.

Tagging $b$-jets is a complex area in High Energy Physics (HEP) that seeks to identify jets that have originated in a bottom-quark [5, 6]. It is customary to construct *b-taggers* [7] that receive the features of a given jet in an event and return a ($b$-tagging) score. This score can be considered as being sampled from different probability distribution functions (PDFs) depending on whether the jet has originated in a bottom-quark (b), charm-quark (c), or light-quark/gluon (u, d, s or g). The less overlap there is in these probability distributions, the better the tagger. This kind of problem is usually faced by defining a working-point as a threshold in this score and then performing the difficult and needed task of calibrating on data the true positive rate for $b$-jets and false positive rates for all other non-b jets whose score results are larger than this threshold.

In this work, we explore the possibility of, instead of using calibrated working points, using the available full score PDFs despite knowing that it should be taken as an approximation of a calibrated curve, which is very difficult to achieve. We develop a new statistical estimation procedure by flexible Bayesian modeling [8] and exploiting structured prior information. Simultaneously, this inference process also returns a posterior for the fraction of events of each class present in the dataset, which is of prime interest in any physics analysis.

The rationale for pursuing our goal is as follows. The object of study consists of multijet events, and therefore the dataset consists of tuples of $b$-tagging scores. Since each one of the events consists of some combination of the individual jet flavors (light, charm and bottom), then the likelihood for a given event needs the PDFs for each flavor. Since the true value for these curves is unknown, but we have some prior knowledge of their shape, we set these curves as parameters in the inference problem. In doing these we show how to take advantage of expected features such as continuity and unimodality to be able to infer arbitrary (nonparametric) continuous curves. The classes in the multijet events are each flavor combination allowed by the physical problem, which is usually a few, and in particular much fewer than all the mathematical possible combinations. Henceforth, as a mixture model problem, the likelihood for one event is the sum of the probability for each class times the probability of the given scores in that class. With the likelihood for the dataset one can then infer a density estimation for the parameters, which are the score PDFs for each flavor and the expected fraction of events for each class.

One of the central problems is the inference of the score PDFs for each jet flavor, or individual components, in the mixture model. In this article, we discretize in bins these curves and we study four modeling strategies. The first is to model these PDFs as being sampled from a Dirichlet distribution a priori; the second is as being sampled from a self-normalized Gaussian process; the third as being a weighted mixture of unimodal curves; and the fourth as a strict unimodal using priors from the previous case. The structured priors in these strategies, which come from shape constraints and/or prior regularization, yield an inner structure to the model that leverages the inference results. Statically speaking, these models share the same likelihood while the difference comes from the

prior. In contrast to the common misunderstanding of Bayesian inference that either the prior does not matter with enough sample size or the prior needs to be avoided for it comes from subjective specification, here we adopt a hierarchical Bayes approach, where the structure of the prior does help extract more information from a limited amount of data, and we are effectively learning the prior using the observed data via the pattern of the smoothness or the shape. When there is a structured shape of the true score PDFs, it is important for the model to use such structure in the estimation. Standard frequentist statistical tools assume exchangeability between bins, or shrink bin height estimates towards a global model in the absence of data. Our structured prior approach is better able to capture much more of the true score PDF shape especially in bins with limited observations.

This work is divided as follows. In Section 2 we write down the mathematical formulation of the model, its parts, and the aforementioned strategies. We first describe a one-dimensional model and then the $D$-dimensional model. In Section 3 we show the results of the paper. First, we show a toy one-dimensional model that only works, and to some extent, in the unimodal models. Then we show the results for a 4-dimensional case and we infer all relevant features with all four models. Section 4 contains discussions of the results and their prospects, and Section 5 details the conclusions of the work. All the results of this article and the corresponding scripts to obtain them, have been placed for downloading and reproduction in the Github repository [9].

# 2 Inferring mixtures of continuous and unimodal distributions

## 2.1 One-dimensional problem

Before the realistic discussion on the multijet events, we start with a one-dimensional task: Given a sample of $N$ independent identically distributed (IID) observations $x_1, \ldots, x_N$, where $x_n \in \mathbb{R}$, we assume the density as a $K$-component mixture.

$$x_n \sim \sum_{k=1}^{K} w_k p_k(\cdot). \tag{1}$$

The model contains three sets of free parameters:

- $K$, the number of mixture components,

- $p_k(\cdot)$, the individual density component,

- $w_k$, the mixture weight.

In the physics problem we consider, each density component corresponds to one jet flavor. For now, we fix $K$ and make inference on $p_k, k = 1, \ldots, K$ and their weights.

If each $p_k(\cdot)$ is specified by a parametric model, $p_k(\cdot|\mu_k)$, the inference is trivial through a maximum likelihood estimate:

$$\max_{\mu, w} \sum_{n=1}^{N} \log \left( \sum_{k=1}^{K} w_k p_k(x_n|\mu_k) \right).$$

But this parametric approach fails in practice for two reasons: First, the parametric model $p_k(\cdot|\mu_k)$ is rarely correctly specified; indeed the shape of the distribution is often of a central interest. Second, due to the discrete nature of the observation, the data we have often comes in a histogram: Without loss of generality, we assume the support of $x$ is compact, $0 < x < M$, and we only measure the binned counts $y_m$: the counts of $x_n$ in an interval $[m, m+1)$, i.e.,

$$y_m = \sum_{n=1}^{N} \mathbb{1}(m - 1 \leq x_n < m), \quad m = 1, \ldots, M$$

.

In this paper, we only consider discretized observations. It is equivalent to viewing $x_n = \lfloor x_n \rfloor$ having support on integers $1, 2, \ldots, M$, and $y_m = \sum_{n=1}^{N} \mathbb{1}(x_n = m)$. Since $x_n$ is assumed to be discrete, each mixture component $p_k(x)$ is a probability mass function on $(1, 2, \ldots, M)$, and is characterized by $\mu_{km} = p_k(x = m)$, the probability of the $m$-th bin in the $k$-th mixture component.

The log likelihood is easy to write by a categorical distribution:

$$\log p(y|\mu, w) = \sum_{m=1}^{M} \left( y_m \log \left( \sum_{k=1}^{K} w_k \mu_{km} \right) \right). \tag{2}$$

We are facing non-identification in the likelihood: We can re-distribute probability masses across mixture components, as long as the quantity $\sum_{k=1}^{K} w_k \mu_{km}$ is invariant. What remains left is the prior specification, which plays a central role in identification. Here we outline four structured prior specifications.

### 2.1.1 Dirichlet model

The individual probability masses $\mu_{km}$ are on a simplex. That is, for any $k$,

$$\mu_{km} \geq 0; \quad \sum_{m=1}^{M} \mu_{km} = 1.$$

The Dirichlet distribution is the easiest prior:

$$\mu_{k1}, \ldots, \mu_{kM} \sim \text{Dirichlet}(\alpha_k), \ k = 1, \ldots, K, \tag{3}$$

where $\alpha_k$ are hyper-parameters that control the concentration of each mixture component. A bigger $\alpha_k$ entails a more spiky probability distribution. In the experiments, we often set $\alpha_k$ to match the shape of a Beta distribution.

### 2.1.2 Gaussian process model

Despite its simplicity, one key drawback of the previous Dirichlet prior is its lack of topological modeling: it treats each bin $\mu_{km}$ as exchangeable. Since the underlying probability density function should be smooth, we want to use the bin locations as extra information, and make partial pooling across nearby bins. To that end, we consider a Gaussian process prior. The first step is to perform a softmax transformation and convert the constrained simplex vectors $\mu$ into an unconstrained parameter $\beta$ :

$$\mu_{km} = \frac{\exp(\beta_{km})}{\sum_{m'=1}^{M} \exp(\beta_{km'})}. \tag{4}$$

We typically set $\beta_{k1} = 0$, and for all remaining $m > 1$, $\beta_{km} \in \mathbb{R}$ will be unconstrained free parameters in the model to learn.

To encode the "distance" of bins, we can adopt a simple exponential kernel Gaussian process prior:

$$\beta_{k\cdot} \sim GP(f(\cdot), \mathcal{K}(\cdot, \cdot)); \quad \text{Cov}(\beta_{km_1}, \beta_{km_2}) = \sigma_k^2 \exp(-\frac{|m_1 - m_2|^2}{2\rho_k^2}), \tag{5}$$

in which $\sigma_k$ and $\rho_k$ are the hyper-parameter controlling the length-scale and scale, encoding how smooth each individual density component is, and can potentially change by $k$. The mean function $f(\cdot)$ is either chosen to be zero, or a given Beta distribution.

### 2.1.3 Unimodal model

Perhaps a more physically meaningful design is to require each individual component to be unimodally shaped. To encode this shape constraint, we still perform the softmax transformation (4). Because this transformation is monotonical, it is sufficient to enforce the vector $(\beta_{k1}, \ldots, \beta_{kM})$ to be unimodal. For each $k$, we introduce a discrete parameter $l_k \in \{1, 2, \ldots, M\}$ that represents the mode of the $k$-th density, and $M - 1$ non-negative increments $\delta_{km} \in \mathbb{R}^+, m = 1, \ldots, M - 1$. First set

$$\beta_{kl} = 0,$$

then for $1 \leq m \leq l - 1$,

$$\beta_{km} = \beta_{k(m+1)} - \delta_{km},$$

and for $l + 1 \leq m \leq M$,

$$\beta_{km} = \beta_{k(m-1)} - \delta_{k(m-1)}.$$

This design ensures that after the transformation, $\mu_{km}$ is unimodal given any realization of $l_k$ and $\delta_{km}$.

The mode $l_k$ can be placed a uniform prior over $\{1, 2, \ldots, M\}$, and the increments can be modeled as independent Gaussian in the prior:

$$\delta_{km} \sim \text{normal}(0, \sigma_k).$$

In the inference phase, we compute the posterior of free parameters $p(\{w_k\}, \{l_k\}, \{\delta_{km}\}|y)$. This task is typically achieved in modern programming languages such as Stan, in which the discrete parameter is handled via marginalization.

### 2.1.4 Point estimate from unimodal prior

The full Bayes outcome from (2.1.3) is a mixture of unimodal densities, integrating over the uncertainty of mode location $l_k$ and increments $\delta_{km}$. Sometimes either for computation cost or for interpretation, we would wish the inferred density functions $\mu_{km}$ to be unimodal. A quick remedy is to first fit the full Bayes model, and pick $\hat{l}_k = \arg\max_m \Pr(l_k = m|y)$, and refit the model using the plug-in estimate $p(\{w_k\}, \{\delta_{km}\}|y, \{\hat{l}_k\})$.

## 2.2 $D$-Dimensional problem

The extension to $D$ dimensions of the above problem consists in having $N$ observations $\mathbf{x}_1, ..., \mathbf{x}_N$, where $\mathbf{x}_n \in \mathbb{R}^D$ and each one of its components corresponds to a one-dimensional variable as defined in Eq. 1. Hence, we extend Eq. 1 to

$$\mathbf{x}_n \sim \sum_{k=1}^{K} w_k \mathbf{p}_k(\cdot), \tag{6}$$

where $\mathbf{p}_k(\cdot)$ is a $D$-dimensional vector whose components, depending on the class, are some specific combination of $p_d(\cdot)$. That is, the classes in the one-dimensional problem become the individual components of the new classes in the $D$-dimensional problem. In this multidimensional problem we number the individual components with $d = 1...D$, and use $k$ to number the $K$ classes; in particular $w_k$ is not the same in Eqs. 1 and 6. The $K$ classes are distinguished by their specific combination of the individual components. The physics of the problem determines which are the combinations of the individual components that are present as the classes in the problem.

Since each class can have a different combination of individual components, then in principle the number of parameters would increase through $w_k$ because of the number of classes $K$. However, being the problem $D$-dimensional, the number of observations increases considerably. If we bin each individual component in $M$ bins, then the observations correspond to a $D$-dimensional histogram with $M^D$ bins. In the physical problem worked out in this paper, the classes are determined by the nature of the problem and correspond to a few specific combinations of the individual components in each class. Therefore the problem becomes identifiable because observations increase more than free parameters.

The likelihood in the $D$-dimensional problem is the sum of the mixture weights for each class, $w_k$, times the product of the individual components PDFs of that class, evaluated at the measured $b$-tagging scores. In the cases of classes in which the individual components are not the same, since the measured scores are not assigned to any component, one has to average over all possible permutations. Therefore, the expression for the log likelihood is

$$\sum_{n=1}^{N} \log \left( \sum_{k=1}^{K} w_k \frac{1}{\text{perm}} \sum_{\text{perm}} \prod_{d=1}^{D} p_d(\text{score}_{i_d}) \right), \tag{7}$$

where $\text{score}_{i_d}$ is the $b$-tagging score of the $d$-th jet in the $i$-th allowed permutation within each class. $p_d(\text{score}_{i_d})$ is the evaluation of the corresponding individual component PDF in the measured score.

The model contains then the same three sets of free parameters as in the previous case (c.f. Eq. 1), with the differentiation that now the classes correspond to $D$-dimensional combinations of the individual components. The model also contains the specification of the combinations in each one of the classes, which in this work is taken as fixed prior knowledge that comes out from the nature of the problem.

# 3 Multijet events with flavor-specific classes

In this Section we implement the tools described above in simplified problems of datasets consisting of one jet or multijet events in which finding out the jet flavor, the event class and the dataset composition is part of the problem. This kind of problem occurs often in collider physics and it can be encountered in LHC searches such as, for instance, $pp \rightarrow t\bar{t}t\bar{t}$, $pp \rightarrow t\bar{t}W$, $pp \rightarrow t\bar{t}h$, $pp \rightarrow hh$ and many others. We describe some of the phenomenology and experimental tools utilized in these problems and explain how they can be matched to be used in scenarios as detailed in Section 2. We then present a simple problem of $N$ events with one jet per event (the one-dimensional problem), and then a richer problem with four jets per event (the D=4-dimensional problem). As discussed in Section 2.2, we find that the richer inner structure of the latter, is more fertile to be exploited with the proposed tools. The problems and solutions presented are a proof of concept, we discuss in Section 4 possible paths to take these ideas to a production level.

## 3.1 Jet flavor tagging

Jet tagging, especially heavy-flavors such as bottom-quark $b$, is an important area in collider physics programs, since its tools and results are used to reconstruct particles such as the top-quark or the Higgs boson. The main objective in this field is to recognize whether an observed QCD jet has originated in a $b$-, $c$-, light-quark, or a gluon. This classification problem has a long history of methods that have been continuously improving over the years. The latest developments consist mainly of a multivariate analysis (usually a neural network) that uses many features in the jet and in the event to assign a score to the jet. Then each kind of jet has a different probability distribution on this score, permitting to filter jets and events in order to perform physics analyses. Creating these tools and algorithms is a sophisticated field driven nowadays mainly by the ATLAS [5] and CMS [6] collaborations at the LHC.

We take as a departure point for our toy-model analysis the $b$-tagging performance results by ATLAS using the GN1 constructed variable [10, 11]. This result, whose relevant features are summarized in Fig. 1, uses Monte Carlo simulations for the hard scattering, radiation, hadronization, showering and the detector to learn a classification algorithm. In order to use the results in Fig. 1 in a physics analysis, a working-point (dashed vertical lines) is defined and it is further calibrated to account for biases and uncertainties. As a matter of fact, the distributions in the figure are expected to have some distortions because of many systematic uncertainties such as for instance tracking efficiencies, pile-up, and many others (see for instance [12]). In addition, one also knows that these distributions depend on the final state. Henceforth, the results in Fig.1 should be taken as an approximated guide of what the real distributions are in data.

For the sake of simplicity, and for the purposes of the proof-of-concept pursued in this work, we take the distribution curves in Fig. 1 as true distributions from which we sample synthetic data[1]. We sample the data from a probability density function determined by these curves; i.e. we do not use any physics Monte Carlo to sample the data. To emulate a scenario in which one does not have access to the true distributions, as it is in real data, we do not use as input information these true curve values at any point in the analysis, except for assessing the algorithm performance. Depending on the model we use as prior knowledge different combinations of the following features: shifted distribution curves (which do not match the true ones, as it is the case in real scenarios), the continuity and smoothness of these curves, and/or their expected unimodality. To avoid label switching along the inference process we also use that their modes have some sorting in the GN1 ($b$-tagging) score. In the case of multijet events we also use as prior knowledge that the jets in one event cannot be of any flavor combination, but only some specific combinations; as it would be in a real analysis where one knows which are the expected backgrounds and their jets flavor combination. We find that this is fertile prior knowledge since it relates the dependence of the GN1 scores in a given event to the information that one pursues to extract from the dataset.

In the following paragraphs we present and solve two problems using the different models and tools described above. In Section 3.2 we address the 1D problem, whose outcome is not good but settles down the proposed idea. In Section 3.3 we solve the problem of four jets per event, whose solution is enhanced by the multidimensionality of the problem, as discussed in Section 2.2.

---

[1]Observe that in the framework proposed in this article we use as true distributions the ones that would be the (surely) biased priors when addressing the real data.

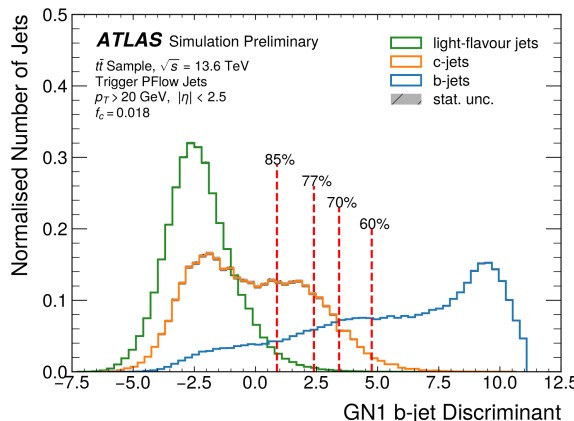

Figure 1: *b*-tagging scores for light-, *c*- and *b*-jets in the ATLAS variable GN1 [10, 11]. Along the model and problem presented in this paper we use the *c*- and *b*-curves from this plot.

## 3.2 1D: One jet per event

Let us suppose that we have $N$ events, each event consisting of one jet whose flavor could be either $c$ or $b$. Here the jet flavor plays the role of the one-dimensional class or individual components in Section 2. We have chosen to work with only two flavors for the sake of simplicity, and we use $c$ and $b$ to make it more challenging because $c$ is the one that looks more alike to $b$ in Fig. 1. The extension to three flavors is straightforward to implement, although its convergence and performance should be correspondingly computed and analyzed. In each event we measure the GN1 score corresponding to the given jet. The objective is to obtain: *i)* the true distributions from which the data has been created; and *ii)* the fraction of events coming from each one of the one-dimensional classes, namely $c$-jets and $b$-jets.

In order to apply the techniques and models described in Section 2 we use a dataset consisting of $N = 500$ events and we divide the GN1 span in 14 bins. We decide to use this number of bins to have a good balance between the smoothness and jaggedness of the histogram for the data [13]. Along this work we indicate each bin with a point in its center and plot lines joining these points. The dataset is created with 80% of its events corresponding to the $c$-jet class and 20% to the $b$-jet class. After binning the data, this one consists of $N = 500$ numbers between 1 and 14. See more details in the Github repository [9].

As a first benchmark model to estimate the densities we use Dirichlet distributions (Section 2.1.1) as priors for the $c$- and $b$-distributions on GN1. We set the Dirichlet $\alpha$ parameters such that their means correspond to beta functions with parameters $\alpha$, $\beta = 4.5$, 10.2 (class $c$) and 4.5, 2.25 (class $b$), which are the dotted lines in the upper row (left and central panels) in Fig. 2. These means are curves that do not match the true curves from which the synthetic data is sampled. The challenge is to determine whether the posterior approaches the true distributions. Observe that this model can only exploit the information coming from the prior means. Since Dirichlet distribution does not provide correlation because of bin proximity, then the model cannot exploit the expected continuity of the curves. This latter can also be seen in the jagged lines in the priors and posteriors for the $c$- and $b$-distributions in the upper panel in Fig. 2. Moreover, as it can be seen in the central panel, the posterior distributions for the $c$- and $b$-distributions do not represent any noticeable improvement with respect to the priors. The relevant reasons are that the Dirichlet distribution is too flexible in neighboring bins and that one jet per event does not contain valuable information on the inner structure of the data. The posterior on the classes fractions in the sample (right panel) approaches the correct value in absolute units, however this is solely because of the prior means which have some similitude to the true curves.

As a second model, we use the Gaussian process model in Section 2.1.2. We use as the mean of the Gaussian the same beta functions as above which are also the dotted lines at the second row (left and central panels) in Fig. 2. The covariance matrix hyperparameters are (along this work) set to $2\rho^2 = 3$ and $\sigma^2 = 0.25$ for both classes (see Section 2.1.2), and we use an `ordered vector` [14, 15] in the fifth bin to fix that in this bin the $c$-distribution is greater than the $b$-distribution. We have also tested inferring $\sigma$ and $\rho$ as parameters, but the running time increases considerably with no

noticeable improvement in the results, and with some difficulties in the convergence of the chains as indicated by the $\hat{R}$ parameter [16] (see Github [9]). Therefore in this model we are using the prior knowledge of *i)* shifted distributions with some proximity to their true values and *ii)* continuity. The results for the inference are plotted in the second row in Fig. 2. We see in the central plot that again there is no noticeable improvement, mainly because of the non-identifiability of the problem. The convergence for the sample fractions has similar results and explanations as the previous case using Dirichlet priors.

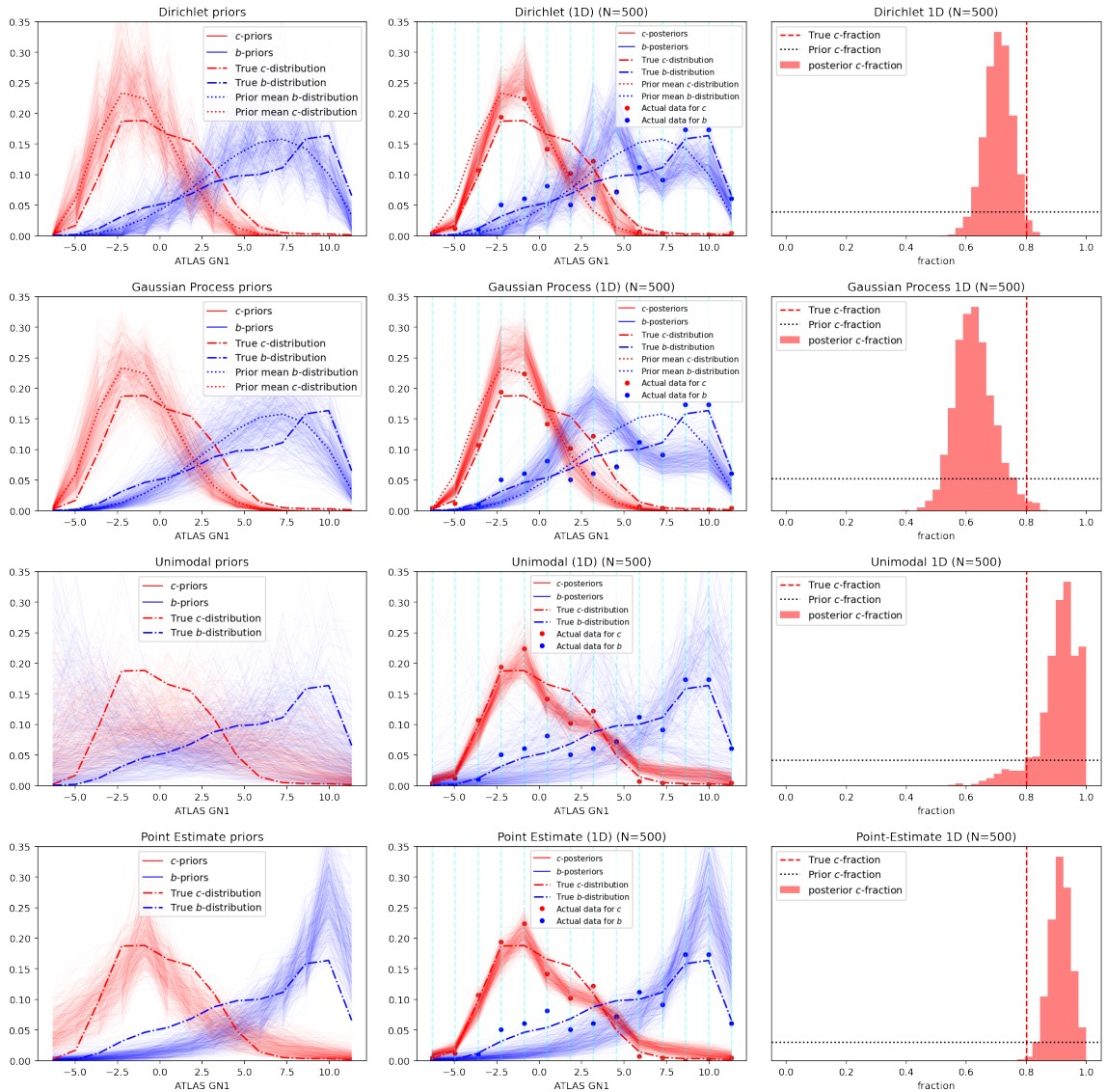

Figure 2: Problem of $N$ events with one jet per event, which can be either from class $c$ or $b$. The problem is addressed with the four described models: (from top to bottom) Dirichlet, Gaussian process, Unimodal and Point-Estimate. The left and center columns indicate the priors and posteriors on the $c$- and $b$-distributions, respectively. The right column shows the prior and posterior for the fractions of events with a $c$-jet. As expected from the discussion on the identifiability in the 1D problem, we find practically null improvement in the Dirichlet and Gaussian process models for recovering the posteriors of the $c$- and $b$-distributions. The Unimodal model, starting from an agnostic prior, recovers a posterior with similar shape to the true curves. As we show in the next Section, this performance it is improved with many jets per event since the dependence between them at the event-by-event level provides profitable information for the inferential process.

As a third benchmark model we use the Unimodal model presented in Section 2.1.3. This model has the virtue that exploits *i)* continuity and *ii)* (mixture of) unimodality, however it is not guided by a prior mean. In any case, we use prior knowledge to also set an `ordered vector` as in the Gaussian process case. This helps to avoid label-switching [17]. We see in this model that the

posterior approaches the true values in comparison to the priors, even in a one-dimensional problem. This is an important achievement and one of the reasons behind it is that unimodal classes break the non-identifiability if the data is not unimodal.

The fourth proposed model is the Point-Estimate (Section 2.1.4), which corresponds to the bottom row in Fig. 2. This model exploits the maximums obtained in the Unimodal model for each class and samples strict unimodal curves with their mode in the most likely mode in the previous case. Since one starts with the prior from the Unimodal model, then there is not much improvement when comparing the posterior to the prior. In any case, there is a fair approach to the true curves, just as in the Unimodal model.

We have used this one-dimensional problem to deploy a simplistic version of the presented framework in a relatively simple scenario. The inference results for the one-dimensional case are not good, except in some cases in the Unimodal instance. This was expected from the discussion in Section 2. In the next Section we analyze the multidimensional case within a framework inspired by a physical problem.

## 3.3  4D: Four jets per event, a toy problem

Having studied the 1D case, we extend to the 4D case in this Section. This corresponds to having four jets per event. As a guide to construct the data we get inspired by the $pp \rightarrow hh \rightarrow b\bar{b}b\bar{b}$ process, where some of the backgrounds are such that in the four jets at the true level there are either 0, 2 or 4 $b$-jets. Although in the real process the backgrounds and the physics are much more sophisticated, the observation that an odd number of $b$'s at the true level is very rare, allows us to consider a simplified model in which the classes can only be *cccc*, *ccbb* and *bbbb*. One of our interests is to obtain the mixture weights for these classes in the dataset. (For our purposes we drop the bar over the quark since we assume equal GN1 distribution for quark and anti-quark jets.) Having only these three classes at the true level yields an important piece of information to be exploited by the models, since now the dependence at the event-by-event level provides profitable knowledge about the inner structure of the data.

In this 4D scenario data corresponding to $N$ events consists of $N$ 4-tuples. We work with $N =$ 100, 250 and 500. We use the same hyperparameters as the 1D case, namely the prior means, $\sigma$ and $\rho$ for the Gaussian process model, and the same hack for the ordered vector. For the sake of comparing these three cases, we use the same number of bins for all the cases. We find that using 24 bins, although it is above the ideal for $N = 100$, works quite well for larger $N$. We can see this just from the data, by analyzing its smoothness and jaggedness, as discussed above. Notice that the 24-bin case yields a considerably better curve resolution than the 14-bin case in the 1D problem.

In the scenario worked out in this Section we have three classes (*cccc*, *ccbb* and *bbbb*), hence the mixture weights distribution in the sample is conveniently plotted in a two-dimensional simplex (from a three-dimensional space). Therefore to show the true fraction, and the prior and posterior distributions for the fractions in the sample, we work with histograms in the simplex. To define these histograms in the simplex we have divided each class fraction into 11 bins. This yields a total of $11 \times 12/2 = 65$ bins, since the three fractions should add to one. We are using a totally agnostic prior knowledge in the fractions, which assigns to each bin the uniform probability of $1/65 \approx 0.015$. We plot colored contour-curves on the simplex based on the counting on these bins. In each simplex plot we indicate how much increases the probability from the prior to the posterior in the true fraction bin.

We show the results for the $c$- and $b$-distributions in a plot with GN1 in the horizontal axis. In these plots we also show how the data was actually sampled in the dataset through red and blue dots (see below in Figs. 3–6). This visualization should be handled with care since each dot represents the fraction of times that any of the jets in the event in the given individual component had the given GN1 score. That is, these dots can only be visualized in synthetic data where one has the true label of each data point. Observe that there is a loss of information because we are projecting into one dimension a four-dimensional result.

### 3.3.1  Results

We have computed the inference for the presented 4D problem using the models proposed in Section 2. In all cases we have started with a given prior knowledge and then computed the posterior after seeing $N = 100$, 250 and 500 events. We have verified the expected and general pattern that increasing $N$

and including more prior knowledge improves the posterior matching to the true parameter values. The results are shown in Figs. 3–6.

Considering the posterior for the $c$- and $b$-distributions in the $b$-tagging score GN1, we can see in the left column in the Figs. 3–6 that the Dirichlet model (Fig. 3) does not have such a good performance as the others in recovering the true distributions. This is mainly because this model does not require these curves to be continuous, which yields variability in the neighboring bins. The Gaussian process model (Fig. 4) incorporates the notion of continuity in the sampling through the off-diagonal terms in the covariance matrix, which provides an important improvement in the posterior similitude to the true curves. The Unimodal model incorporates continuity and (Dirichlet-weighted mixture of) unimodal curves, which it can be expected when using physically meaningful variables to distinguish the classes. In the analyzed case we use the outcome of the GN1 Neural Network, where the unimodality is slightly compromised for the $c$-curve. We further discuss this point in the next section. Finally, the Point Estimate model is tailored to sample strictly unimodal curves, as can be seen from the priors in the upper row in Fig. 6. Observe the considerable improvement for these models from the 1D case in Fig. 2 to the presented 4D case, in which the dependence of the multidimensional data at the event-by-event level provides crucial information to go from the priors (upper rows in Figs. 3–6) to the posteriors (other rows in the same figures). These improvements can be seen at the individual component level (left row in the aforementioned figures) and at the fraction level (right row in the same figures). We find that the Gaussian process, Unimodal and Point Estimate models have an approximately equal level of convergence in the problem studied in this article.

It is worth observing that these results show that even if the prior knowledge on the $c$- and $b$-distributions does not correspond to their true distributions, the inference process retrieves the correct curves and fraction within the corresponding uncertainty.

To quantify and analyze the convergence to the true $c$- and $b$-distributions in GN1 for all models and also 1D and 4D scenarios, we have summarized the results in Fig. 7. For each run, we plot on the vertical axis the root mean square (RMS) distance between the true curve and the posterior samples averaged in all bins and classes. On the horizontal axis we compute the density on the true curve averaged for all bins and classes, plotting the exponential of the mean of the logarithm of this probability, see Ref. [9] for details. As it can be seen from Fig. 7, the posterior convergence improves as $N$ increases and as we use models that include more prior knowledge; in this case the continuity. This plot summarizes and quantifies the posterior convergence for the $c$- and $b$-distributions for all models and scenarios.

Considering the posterior for the fractions of each one of the classes *cccc*, *ccbb* and *bbbb*, we should read the right column in Figs. 3–6. We can see that in all cases the posterior distribution for the fractions is close to the true value in absolute distance, which is a good indicator. If one analyzes how it changes the probability in the true bin from the prior to the posterior, we also see that in all cases it increases. This quantifies the improvement in the posterior probability compared to the prior probability in the true fraction bin. We collect all these results in Fig. 8. Despite some statistical fluctuations, we also observe a pattern that corresponds to more probability in the true fraction as one increases the number of events.

To bolster the presented results we show in Fig.9 the same summary plots for a few other runs using different seeds. We find that essentially the same pattern is observed. All the presented runs have specific seeds and can be reproduced with the information and scripts in [9].

In Fig. 10 we perform a light statistical analysis to compare the model behavior in retrieving the correct fraction for the classes in the dataset. We do not include the Point Estimate model since its behavior is very similar to Unimodal, from which it is derived. We study in the left panel of this figure the model robustness to avoid label-switching, and in the right panel its robustness to increase the posterior density in the true bin. We find that the Unimodal model may lay its posterior mean shifted from the true fraction (left panel), however its increase in probability at the true bin is as good as the other models (right panel). Dirichlet and Gaussian process models have a good centering of their posteriors around the true fraction and mostly without label-switching.

## 4    Discussion

The ideas presented in the previous sections are a novel application in High Energy Physics using and adapting tools from the Statistics and Computer Science fields. As such, they have prospects

to be further developed within the presented framework as well as applied in different areas of High Energy Physics or others. We discuss these prospects in the following paragraphs.

The first observation is on some of the results in the one-dimensional problem in Section 2 (Fig. 2). We have found that if the data is not unimodal, then the Unimodal model can recover some aspects of the mixture distributions, even without the multidimensionality leverage. However, it does not have enough information to extract the details from the true $c$- and $b$-distributions in a univocal manner, as expected. In any case, this inference that starts from agnostic priors it could be potentially improved. It would be interesting to further develop and understand in which cases one could use the unimodality leverage to extract better details from the mixture model. This could include using more prior information on the curve shapes, using more physically meaningful variables to have smoother curves, and including this in the modeling accordingly, among other techniques that should be investigated.

The results in this article are connected to the results in Ref. [18], in which it is shown how Bayesian techniques generalize and improve some customary HEP tools for data analysis. In the present work, we have extended the aforementioned analysis by generalizing that the jets $b$-tagging score probability distribution can have arbitrary continuous, preferably unimodal, distributions. This result implies an important approach to the application of the proposed method in realistic analyses.

The current article, as Reference [18], are both building blocks that aim to construct a different approach to measure $pp \to hh \to b\bar{b}b\bar{b}$ by fully exploiting the available information in the data. Of course, this is a large enterprise that needs many building blocks. Further improvements are in the pipeline and include the treatment of some of the systematic uncertainties that may yield dependence among the in-principle conditionally independent observables. Progress in including systematic correlations in a mixture model can be found in Ref. [19], whose scheme could be transferable to this case. For a realistic analysis, all systematic uncertainties need to be studied, understood and modeled, and all relevant backgrounds included in the model. The most challenging background is non-resonant $b\bar{b}b\bar{b}$, because of its irreducibility in the jets flavor. Observe that despite that in current ATLAS [20] and CMS [21] $pp \to hh \to b\bar{b}b\bar{b}$ analyses this is the only main background, within the proposed Bayesian framework more backgrounds could play an important role in the analysis. This is because the proposed Bayesian techniques are such that many signal events that are tagged as $3b$ in the non-Bayesian framework, will enter into the analysis because of the soft-assignment enhancement (see discussion in Ref. [18]) that allows the inclusion of more signal events in the analysis.

One of the achievements of the study presented in this work is to extract the correct individual $c$- and $b$-distributions. However, in the state-of-the-art of $b$-tagging techniques, these distributions are not physically meaningful, but instead an output from a multivariate classifier. Moreover, our developed tools include leverage for unimodal distributions, something more expected from a physically meaningful variable that could be used to distinguish the individual components. Henceforth, we find that the proposed algorithms may be favored if instead of using a multivariate classifier one uses more meaningful variables, such as Secondary Vertex (SV) taggers, Impact Parameters (IP2D, IP3D) [22, 23], soft-lepton taggers [23] or some combination of physical variables. Such a project would be further favored by the fact that all the distributions for the physical variables can be simultaneously inferred at once using Bayesian inference. In contrast to current analyses in which the $b$-tagging performance is determined separately to the $pp \to hh \to b\bar{b}b\bar{b}$ analysis.

Using more than one physically meaningful variable to distinguish the individual components does not represent a large complication in the Gaussian processes techniques, where one can implement Gaussian Random Fields. However, it may yield some difficulties when attempting to model and sample prior unimodal distributions in more than one dimension. This should be studied in more detail in future works.

Finally, and in general, one should envisage that the presented techniques that yield good inference in mixture models with arbitrary, continuous and preferably unimodal distributions, could play important enhancements in other areas. For instance, in observables where QCD corrections play an important role, one could start with the current order computation of some given distributions, and let the algorithm infer the true distributions for each class by seeing the data. Such a project on QCD corrections could be validated by starting with LO distributions and verifying whether the inference approaches the NLO or further order available distributions. Such a work would be favored if the process has independent observables that can provide multidimensionality to the problem.

# 5    Conclusions

We have implemented a Bayesian framework to extract distributions and fractions in a mixture model of multijet events with different flavor combinations, at collider data. We have shown that, even if the $b$-tagging score distribution for each flavor is unknown and arbitrary, one can exploit its expected continuity and in some cases unimodality to infer its true distribution. This inference needs more than one jet per event to exploit the leverage provided by the multidimensionality of the problem, and it is also notably favored by the prior knowledge of the jet flavors expected in each class. The method also extracts the correct fraction of each class in the sample. Being both, the distributions and the fractions, variables of interest in a physics analysis.

We have implemented four models to infer binned $b$-tagging score distributions and the fraction of each class in a dataset consisting of $N = 100$, 250 and 500 events with four jets per event. The allowed classes correspond to the *cccc*, *ccbb* and *bbbb* flavor combinations. In a Dirichlet model, we find that the lack of correlation in neighboring bins negatively affects the inference of the $c$- and $b$-distributions, whereas the inference for the fraction yields a result relatively close to the true fraction value. In a Gaussian process model, where continuity is exploited through the multivariate normal covariance matrix, we find that $c$- and $b$-distributions and the fraction are correctly extracted through the presented algorithm. We have developed a Unimodal model that samples a mixture of strictly unimodals weighted through a Dirichlet with small parameters, which yields mostly unimodal distributions. This model also correctly extracts the $c$- and $b$-distributions and the fraction. Using the Unimodal model as prior we have defined a Point-Estimate model which is strictly unimodal and also performs correct extraction of the relevant distributions and fraction in the problem. We summarize the performance of the models in Figs. 7 and 8, and we also show the summarized results for other seeds in Fig. 9. We find in all cases the same general pattern: inference works correctly as described above, and the posterior for the individual distributions and the fraction approaches the true value as the number of events $N$ increases. We stress that this is achieved by starting with a biased or agnostic prior and just seeing the data with the correct model for it, demonstrating that the method could be robust to extracting the correct distributions and fractions in a real scenario in which one also starts with a prior knowledge that in general does not match the true values. This result is useful to reduce the impact of Monte Carlo simulations, as well as to eventually reduce the need for calibrated $b$-tagging working points.

We have discussed which points should be studied to apply the present analysis in a realistic scenario. Among the most important we find a systematic potential dependence between the jets scores in the same event, as well as understanding and modeling systematic uncertainties in general. We also discussed how the present results could contribute to improving the measurement of $pp \rightarrow hh \rightarrow b\bar{b}b\bar{b}$, and some other prospects in utilizing the proposed techniques.

We find that the presented algorithms and especially the Unimodal model could enhance its performance if instead of using a multivariate result to perform $b$-tagging, one uses more physically meaningful variables, such as for instance Secondary Vertex or Impact Parameter taggers. In this case not only the unimodality in the distributions would be better established, but also one could work out together and simultaneously the inference for the tagging algorithm and the mixture model, improving the overall algorithm performance. Moreover, in this case the inference on the distributions would be on a physically meaningful variable and therefore the resulting posterior $c$- and $b$-distributions would have a physical meaning that could also be exploited. This will be studied in a future project.

The obtained results suggest that multijet physics analyses could be improved by exploiting information in the data and different aspects of prior knowledge that have been previously disregarded. We have shown that this is the case within some simplifications and approximations, and we conclude that further work to approach more realistic scenarios could achieve more robust results which could be applied in a real data analysis being competitive with current LHC physics analyses.

## Acknowledgments

We thank Manuel Szewc for fruitful discussions and contributions in the early stages of this project. The idea behind this work was born and discussed in the *Voyages Beyond the SM* (V[th] edition) workshop, we thank all participants for the enhancing debates around this and all other ideas. We thank A. Gelman, A. Schwartzman and R. Piegaia for useful discussions.

# A  Computational considerations

Along the presented benchmark problems and models, we have used Hamiltonian Monte Carlo (HMC) through the library `Stan` [24] to numerically compute posteriors in each presented framework. All the required scripts are available in the Github repository [9]. In any case, a few details are pointed out in this section.

We have used the `Stan` command line version `cmdstan 2.33.1`, which has robust documentation and allows a straightforward interaction with the kernel. We have used a commercial desktop computer with 20 threads `i9` and a RAM memory of 64 GB. We have not used GPU.

In most of the cases we have used four independent chains to run the inferential problem, with 1k samples to warm up and 1k samples for the posterior. Our most important variable to diagnose the inference is $\hat{R}$ [16] whose convergence to one determines the good mixture of the chains and defines an unbiased posterior. In most cases we obtain $\hat{R} < 1.02$, and in a few cases $\hat{R} \approx 1.05$.

To optimize the running time and fully exploit the available threads, we have programmed parallel computation within each chain. The details are found in each of the `.stan` scripts in Ref. [9].

Running times in the aforementioned commercial desktop are as follows. For the Dirichlet, Unimodal and Point-Estimate model, running time ranges between 2 to 20 minutes for $N = 100$, 250 and 500. On the other hand, the Gaussian process model runs have taken from one to two days, with time increasing from $N = 100$ to $N = 500$ events, because of the resources consumed of multivariate normal sampling in many dimensions.

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

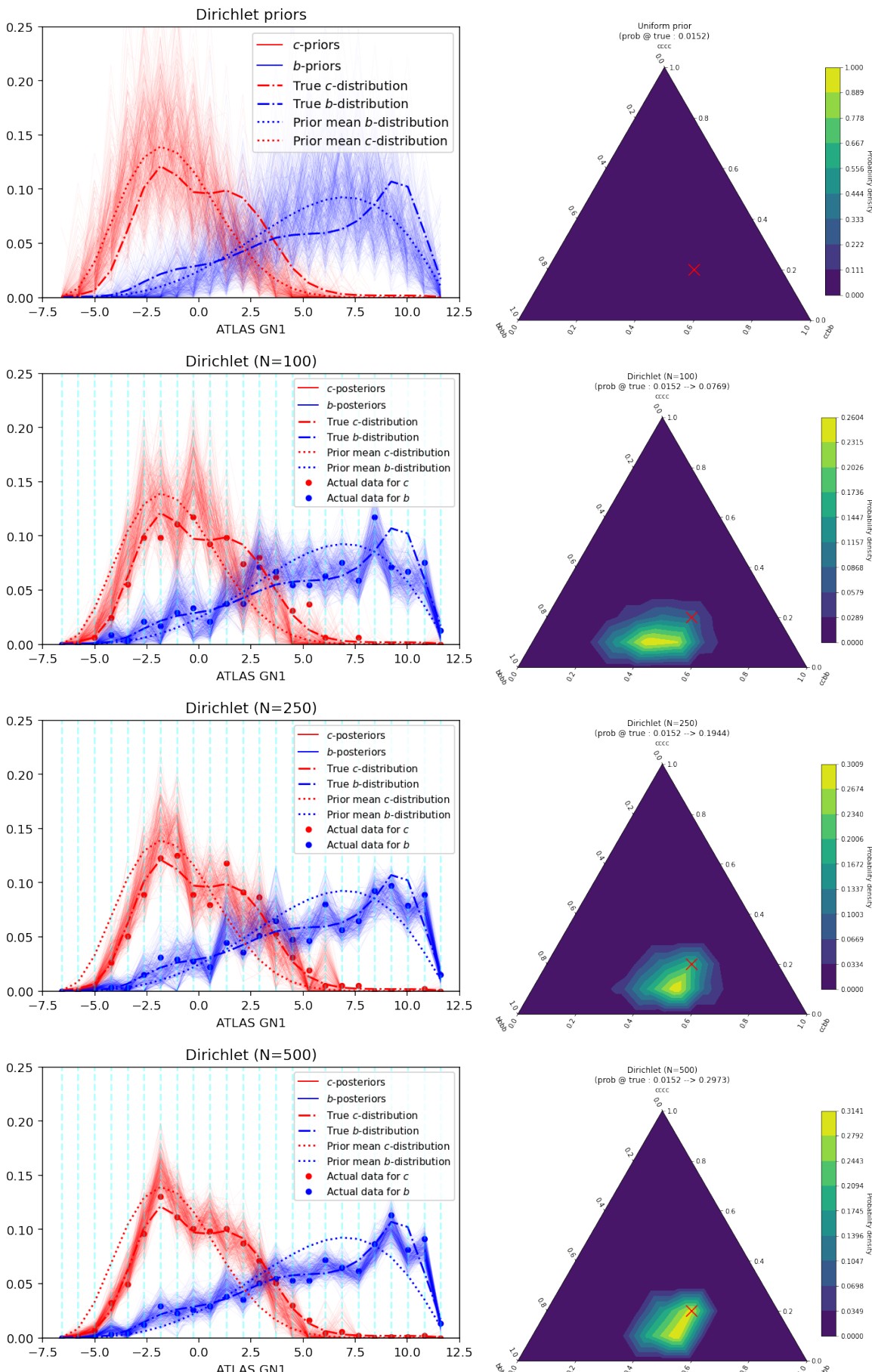

Figure 3: Problem of $N$ events with four jets per event, which can be from either of the classes *cccc*, *ccbb* or *bbbb*. Figure shows the learning stages using a Dirichlet model. The upper row corresponds to the prior knowledge on the *c*- and *b*-distributions (left), and uniform mixture weights of each one of the classes in the sample (right). The red cross in the simplex indicates the true value for the class mixture weights. The following rows correspond to the posterior knowledge of the same properties after having seen $N = 100$, 250 and 500 events, respectively. See text for the *'Actual data'* and for the binning in the simplex plots.

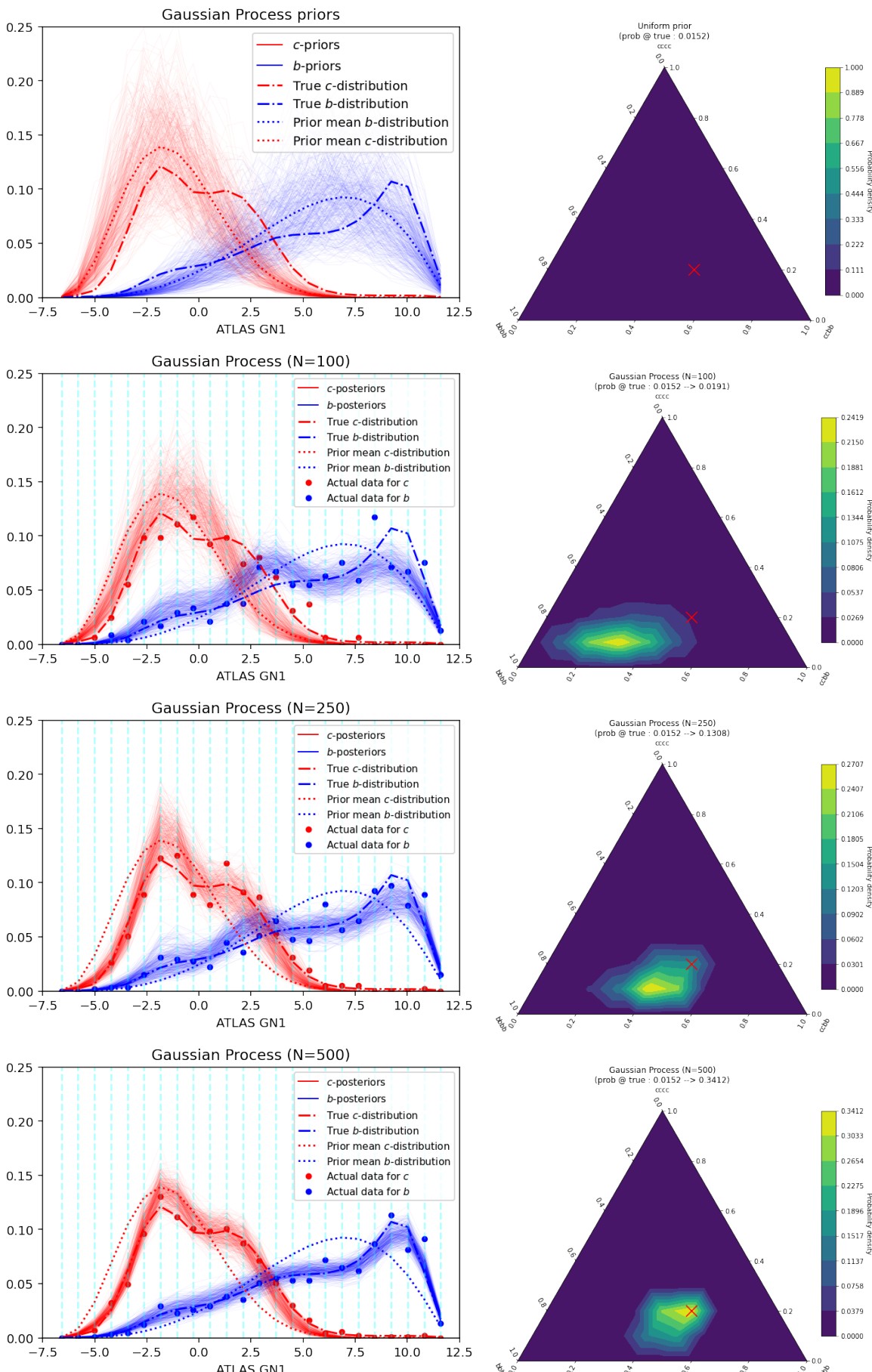

Figure 4: Learning flavor mixtures for Gaussian process model. Same details as in Fig. 3.

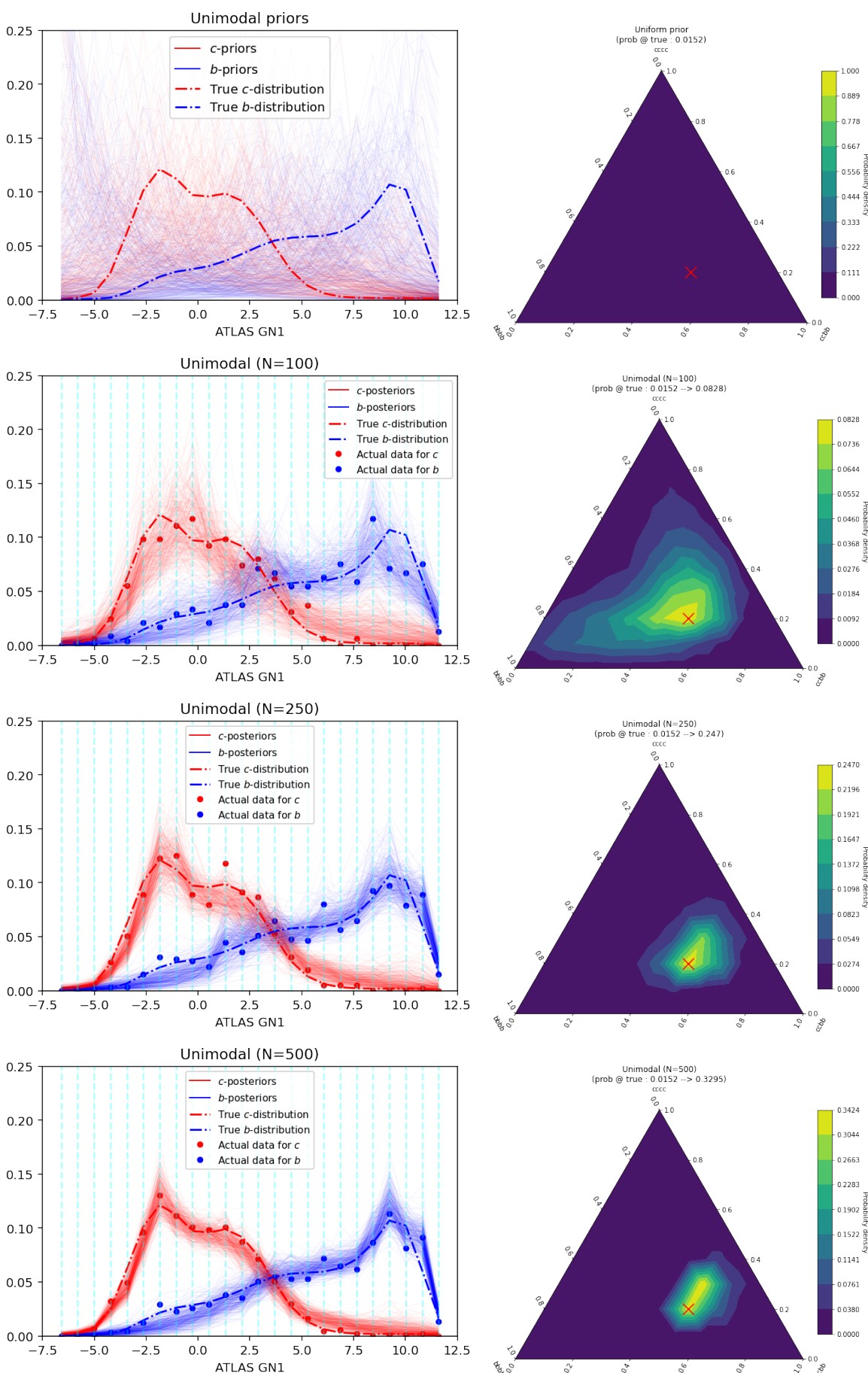

Figure 5: Learning flavor mixtures for Unimodal model. Same details as in Fig. 3.

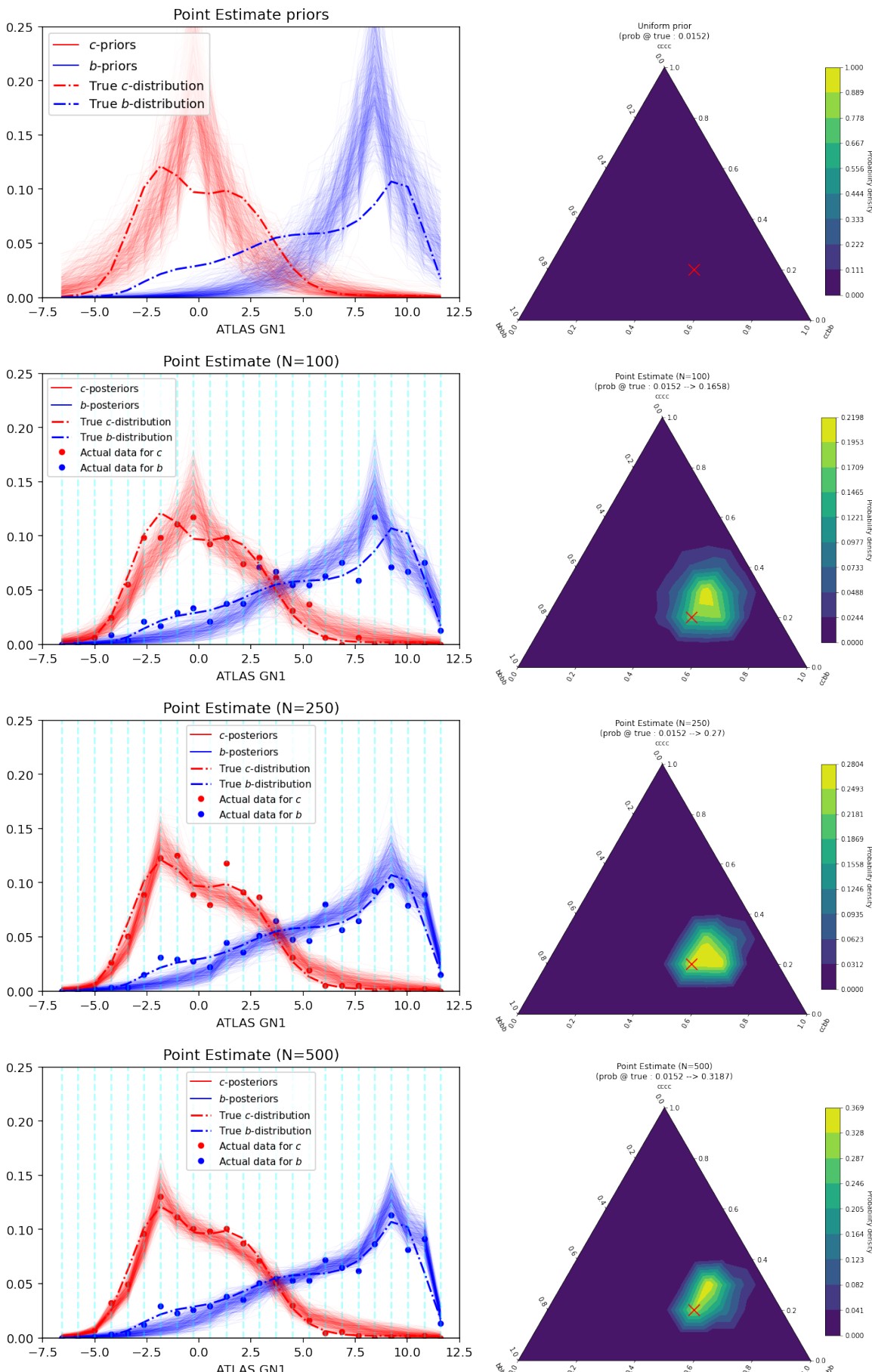

Figure 6: Learning flavor mixtures for Point Estimate model. Same details as in Fig. 3.

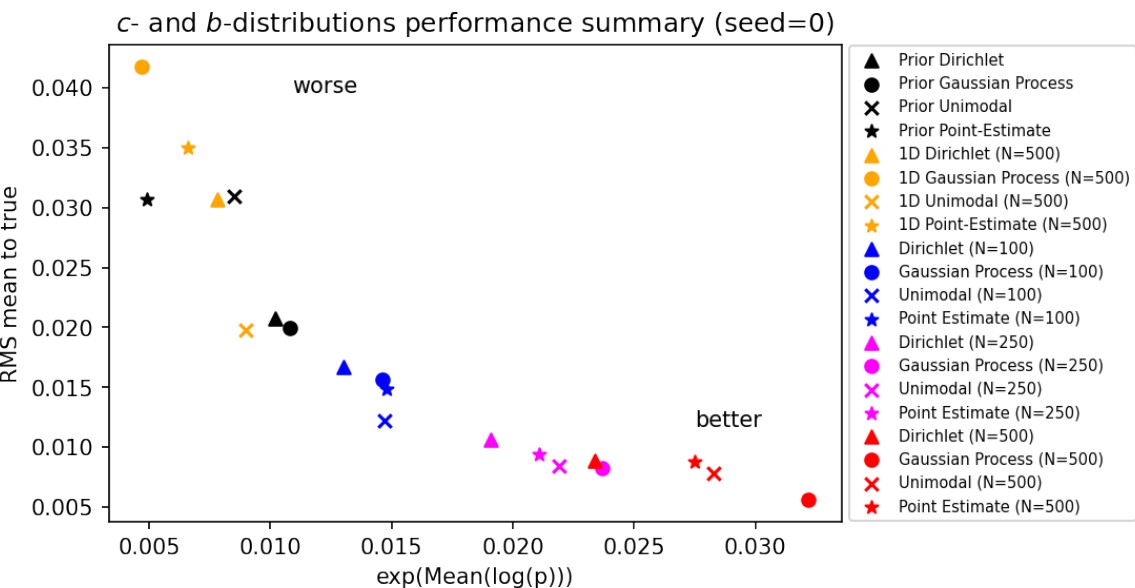

Figure 7: Summary plot for all models and scenarios posterior convergence on $c$- and $b$-distributions in GN1. See text for details.

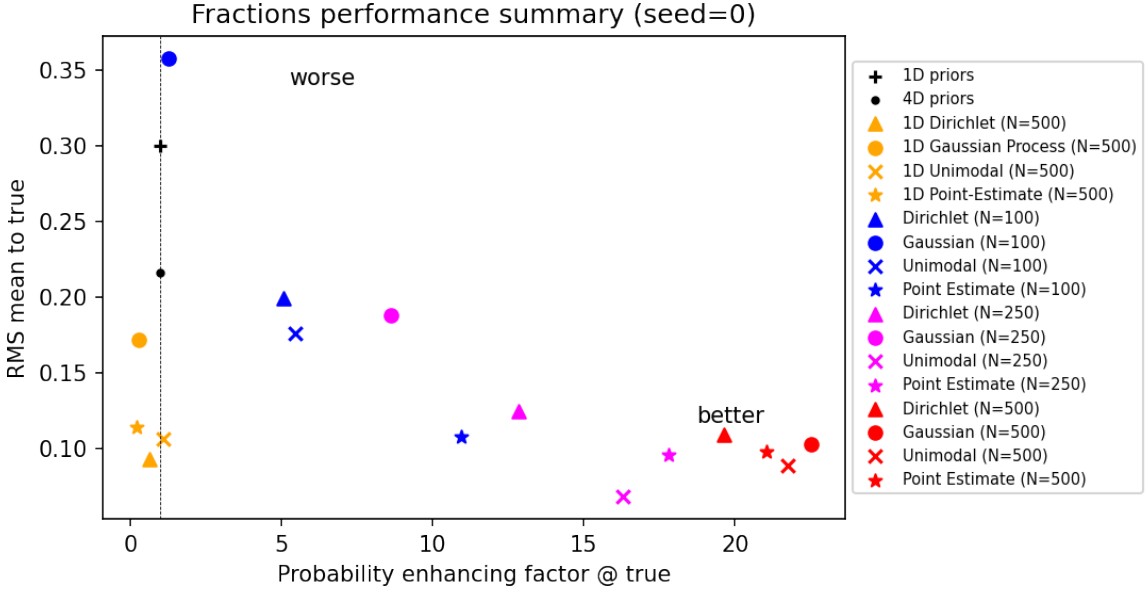

Figure 8: Summary plot for all models and scenarios posterior convergence to the classes fraction in the samples. Observe that in contrast to Fig. 7, in this plot each point refers to only one parameter (the classes fraction) and not an average over many (each bin in the $c$- and $b$-distributions), therefore we should expect a more fluctuating plot. In any case, it can be recognized as a pattern in which the performance improves with data ($N$) and by including more prior knowledge in the models. Here the horizontal axis indicates the ratio of posterior to prior probabilities in the bin where lies the true fraction. See text for details.

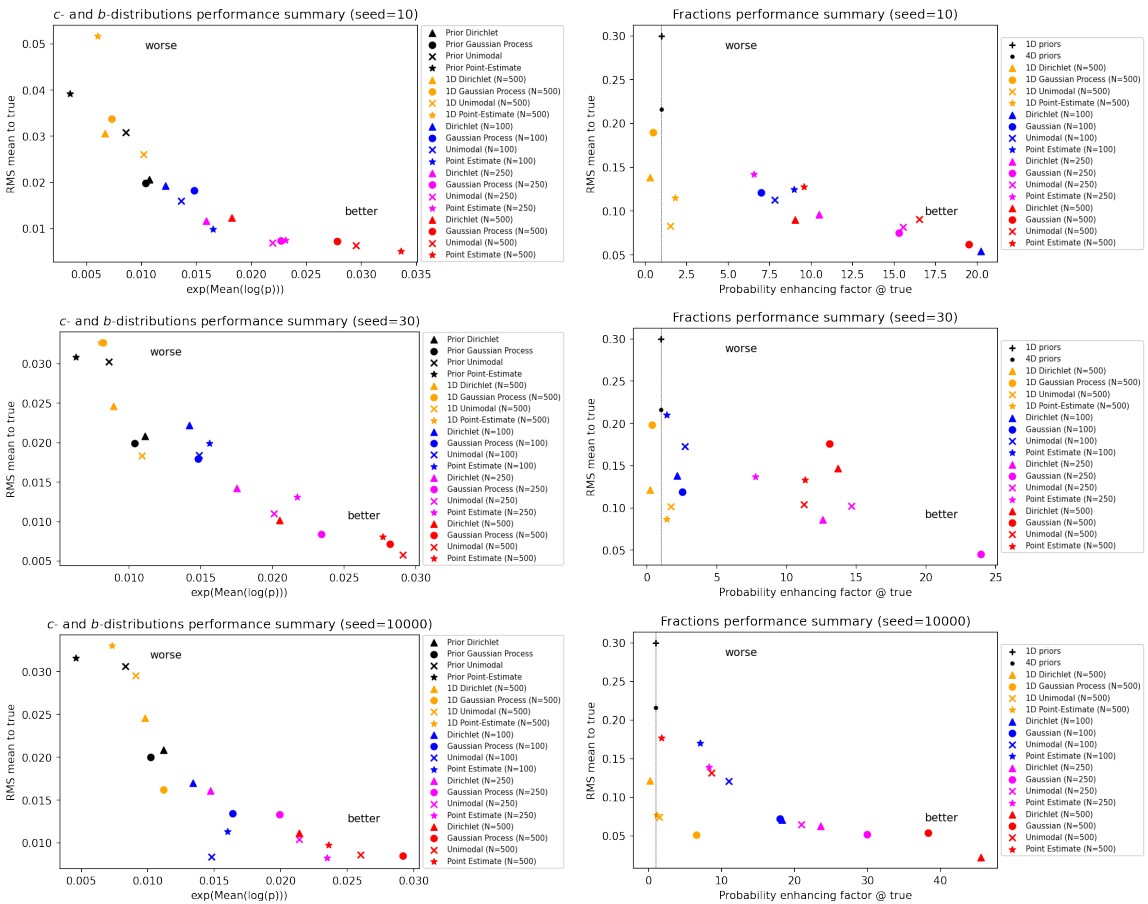

Figure 9: Summary results in distributions and fractions for three different random seeds. Plots are similar to Fig. 7 and 8, but for different data realizations because of changing the random seed. We observe the same general pattern.

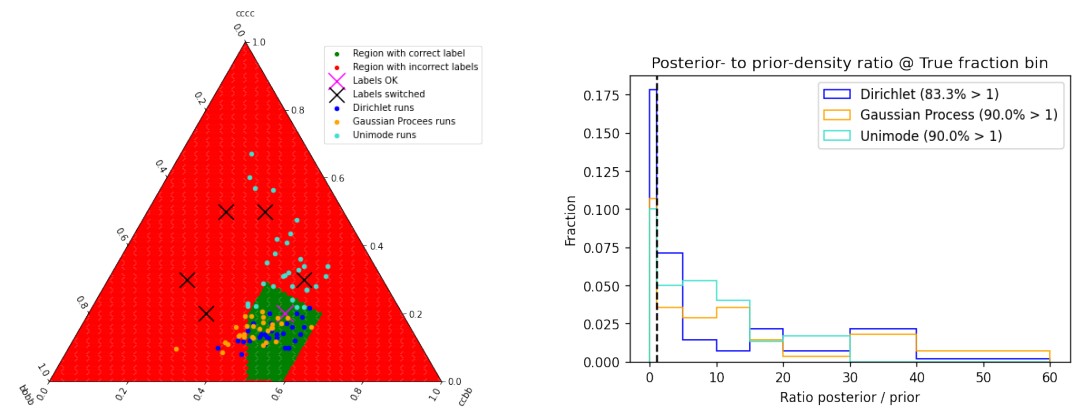

Figure 10: *Left:* Mean fractions for 30 runs of $N = 250$ events using Dirichlet, Gaussian-process and Unimodal models. We plot in magenta cross the correct mixture fraction and in black crosses the other five label switching points in the simplex. The green region shows the area in which each class fraction is closest to its true fraction. *Right:* Distribution of the ratio of densities from posterior to prior in the correct fraction bin (at the magenta cross in the left plot) for all 30 runs in each model. We see that the Dirichlet and Gaussian process are the models whose fractions means are closer to the correct fraction (left plot), however, all models yield an approximately similar improvement for the probability in the correct fraction bin (right plot).