# Peer review of "Inferring flavor mixtures in multijet events"

_SciPost Physics_

## Round 2 · Referee Report · Anonymous (Referee 1) · 2024-6-11

Strengths

  1. The paper presents a novel statistical approach for going beyond fixed working-point jet flavour taggers when analyzing multi-component and multi-jet event datasets.

  2. The ideas presented in this work have multiple potential generalizations and applications to other similar problems in high energy particle phenomenology.

Weaknesses

  1. The methods are tested on toy examples and without direct comparison to existing established approaches. Consequently, their performance and viability in realistic scenarios is difficult to evaluate.

Report

In the submitted manuscript, the authors explore a bayesian statistical method to infer mixtures of processes producing events with multiple quark-flavoured hadronic jets without relying on fixed working point jet-taggers. The method aims to extract the weights of the process components as well as the jet discrimant performances in-situ. It leverages the co-occurrances of different jet flavour combinations, the knowledge of possible truth-level flavour multiplocities predicted by individual component processes, as well as general assumptions on the smoothness and unmodality of physical jet flavour discriminants. The later are implemented through a choice of different Bayesian priors leading to several distinct statistical models with varying prediction and computational performances.
The presented research is topical and potentially highly relevant for experimental collaborations at the LHC. The approach also has potential to be developed further and applied to other similar datasets using continuous discriminants as part of object/event selection.
However in its present form, the paper fails to convincingly demonstrate the full potential of the method as well as pinpoint all of its potential weaknesses. Thus I would ask the authors to fully address the issues listed below before making my recommendation.

Requested changes

  1. The authors should quantify the inference power of their method by comparing it to a more traditional cut-based approach.

  2. In the toy example the "truth-level" fractions of the two process admixtures are both reasonably large. In practice however, one often deals with signal to background ratios which are orders of magnitude below unity (before applying jet flavour discrimination). It is not clear how the proposed method would perform in this kind of scenarios.

  3. From the existing discussion it is also not clear how the different models' computational and discriminative performance scales with the dimensionality of the problem, both in terms of the number of jets, number of events and number of admixtures. Thus it is difficult to evaluate its applicability to other, real world scenarios.

  4. More specifically on the last point: From figures 3 & 4 it seems that for low number of events (N=100), the posterior distributions of admixture fractions are significantly inconsistent with their true values, thus exhibiting bias and potentially putting the method's robustness in question.

  5. In their current approach, the authors choose to bin the otherwise continuous discriminant distribution to build tractable statistical models. It is not clear how essential this simplification is and if there are ways to model the full continuous discriminant response (perhaps using techniques similar to PDF extraction by the NNPDF collaboration).

  6. The labeling of true and prior c-fractions in Figure 2 are confusing. Why is the later drawn as a horizontal line? Shouldn't it be a (flat?) distribution? Then I would suggest a different visualization, to make this more obvious.

  7. The text contains several typographical and grammatical errors as well some possibly incorrect statements. Some examples to be revised a listed below:

a) Second sentence on page 2 is over the top. To be removed or revised.

b) Page 2, line 9: expression "mutual information" is not clear. It may or may not refer to the formal, technical term. To be revised.

c) Page 2, paragraph 3, first line: Acronym HEP is never used. To be removed.

d) Page 9, 2nd paragraph of Sec. 3.3, line 3: the term "hack" is inappropriate in this context. To be revised.

d) Page 9, 3rd paragraph of Sec. 3.3: the last sentence is grammatically incorrect and should be revised.

e) Page 10, last paragraph above Sec. 4: the expression "light statistical analysis" is vague. It should be revised and made more precise.

f) Page 10, last paragraph above Sec. 4. The meaning of the next to last sentence starting with "We find that the Unimodal..." is unclear. It should be revised.

Recommendation

Ask for major revision

  • validity: ok
  • significance: high
  • originality: high
  • clarity: good
  • formatting: excellent
  • grammar: acceptable

Author:  Ezequiel Alvarez  on 2024-09-11  [id 4767]

(in reply to Report 2 on 2024-06-11)

Dear Editor,

We thank the Referee for reading the manuscript and their valuable comments, which have considerably improved the manuscript.

Please find full response in attached document. New version has been submitted to Arxiv, to appear on September 13th.

Ezequiel and Yuling

Attachment:

Referee_2.pdf

---

## Round 2 · Referee Report · Anonymous (Referee 2) · 2024-6-13

Strengths

  1. It is always interesting to see a new idea to solve long-standing challenges in LHC physics;
  2. Combining distributions and fractions for several jets at a time might be useful for well-separated jets;
  3. The papers comes with code, so these methods can be tested.

Weaknesses

  1. I am missing a well-defined case for a combined analysis of many jets, this is assumed, but the reality is much more complicated than even numbers of heavy flavors; 2.

Report

I generally like new ideas for long-standing problems, and the paper presents a new, global approach to flavor tagging. Learning fractions and distributions at the same time is also not unrealistic, and the paper nicely illustrates how the problem can be tackled using a set of standard numerical approaches. In that sense, the paper is interesting and could eventually be published.

However, I see a series of shortcomings: 1. It would be nice to have an actual use case, the reality of jet tagging in events is much more complicated than an even number of heavy flavors; 2. Following a proper use case, the results could also be presented in a physics context. For a journal like SciPost I do not consider pure toy applications without physics interpretation appropriate; 3. In Sec.2 the four different approaches are defined, but it is not clear how they are selected and why these ansatzes have been chosen, please clarify and maybe also specify expected strengths and weaknesses motivating this selection; 3. The rest of the paper is very text-heavy and hard to read. It would help to maybe split up the figures and align them with the text, so one does not always have to flip to the back of the paper. 4. I am not sure if I am missing something, but it would be nice to discuss in more detail the impact of the prior and how this Bayesian approach works in Sec.3; 5. Along the same lines, is the Bayesian approach guaranteed to converge or lead to a stable result? It is not clear to me if I should deduce this from the shown results; 6. So what is the bottom line? Does the method work? Better than what? Or does it need additional work? The paper leaves the reader a little in the dark. Altogether, I think the paper needs a more realistic use case/example/bottom line to be really helpful as a physics paper. The idea is very exciting, though, and I am looking forward to some more quantitative results.

Recommendation

Ask for major revision

  • validity: -
  • significance: -
  • originality: -
  • clarity: -
  • formatting: -
  • grammar: -

Author:  Ezequiel Alvarez  on 2024-09-11  [id 4766]

(in reply to Report 3 on 2024-06-13)

Dear Editor,

We thank the Referee for reading the manuscript and their valuable comments, which have considerably improved the manuscript.

Please find full response in attached document. New version has been submitted to Arxiv, to appear on September 13th.

Ezequiel and Yuling

Attachment:

Referee_3.pdf

---

## Round 2 · Referee Report · Anonymous (Referee 3) · 2024-6-16

Strengths

  1. Well written and presented
  2. Toy application of a less used Bayesian statistical method to a HEP problem

Weaknesses

  1. A bit too much assumption of familiarity with statistical issues and distributions for a physics audience
  2. Lack of clarity on generation and inference methodology, and on which physics information makes the problem tractable
  3. Does not go beyond toy studies based on sampling from 1D curves which could have come from any topic area: the coupling to HEP is rather shallow (other than in limiting the multivariate mixtures to assume pair production of heavy flavour)

Report

A nice demonstration of Bayesian mixture methods to toy problems of tagging-score distribution inference, using four different statistical models. Convergence is shown to require additional physics information such as correlations between multiple jets in an event (this is in need of some clarification) or knowledge of e.g. smoothness and unimodality of the posterior distributions. The amount of physics in the paper is rather small, being more a context for statistical application than central to the conclusions as currently written. No physics simulation is explicitly used, instead simply sampling from provided 1D distributions, and inferring them as a closure test. On the basis that the connection between the physics content and the statistics content is rather small, I think this does not fulfil the SciPost Physics criteria of novelty and synergy between the two fields, though it is a useful demonstration of statistical methods to a physics audience: Physics Core is more appropriate on the declared conditions. It is well written and presented, and a solid contribution to the literature, but does need clearer explanation and perhaps deeper study of the additional information that makes inference possible in the 4-jet case.

  • In abstract and introduction, it would be helpful to clarify that "heavy flavour" in this work refers to b-jets and c-jets but not top quarks -- which are heavy flavour but kinematically distinct and cannot usually be associated to a single jet.

  • "Statically" -> "Statistically" at the end of p2

  • Sub-sections 2.1.x: it would be useful to have more explanation of the distributions' features and motivations, for accessibility to physicists, and more motivation/discussion of the 4 options' pros/cons from a physics point of view

  • Sec 3 start: "In this Section" -> "In this section" (not used as a proper noun here)

  • Sec 3.1: it would be good to note that the identification of a hadronic jet with a specific parton flavour is not a complete picture -- as that is equating a colour-singlet object with a non-singlet and the QCD radiation structure of jets emerges from colour dipoles between partons -- but that it has served well as an approximation, particularly for heavy-quark tagging and to a much lesser extent for quark/gluon discrimination

  • Sec 3.1: LHC collaborations are also increasingly using continuous or pseudo-continuous b-tagging, in which scores are used directly or in differential binnings rather than below/above thresholds for fixing working points. This introduces challenges of calibration, but of course provides more nuanced information for analyses.

  • p6: "label switching along the inference process" needs some clarification: this whole sentence seems a bit cryptic to a typical physicist

  • Sec 3.2: an explicit statement of the computing setup used for these modelling studies would be useful. Stan has been implied but is not explicitly stated, with references to e.g. the ordered vector use appearing in "computing font" but without enough context to understand it. Equally, samples sizes, etc. would be useful to have stated. (I see the computing is specified in Appendix A. Either bring it forward into the main text -- I think think would be best -- or explicitly instruct the reader to see App A for details and avoid having the main text refer to features that require the computing context.

  • Sec 3.2: why are these choices of GP hyperparams the appropriate/natural ones?

  • Sec 3.2, p8: how is it known that the reason for non-improvement in the GP case is "non-identifiability of the problem"? And what exactly does this jargon mean... that the absence of other variables with which to correlate means there is not enough information to unmix the distributions?

  • p8: the b distribution seems to have semi-converged, but in the opposite direction of movement from the prior than what was required. (The Dirichlet case also has a bit of this -- less obviously converged, but with similar features.) What's the reason for that? Can this be sort of mismatch be predicted from the shapes of mixture functions that are being closure-tested?

  • p8: unimodality does not seem fully guaranteed: both the b and c distributions in Fig 1 have inflections on their tails that come close to, or marginally drift into, multi-modality. In general is there a necessity that tagging score distributions will be unimodal when transported to a process other than that where they were trained/calibrated? For example, calibrations from ttbar events might well generate additional structures when applied to jets in events with more tops.

  • p8: it would be good to float Fig 2 to the [b]ottom of the page, to avoid the bit of text on model 3 getting "trapped" there, and easily missed by the reader

  • p8/9: it's claimed that the unimodal mixture model posterior "approaches the true values" by comparison to the priors (which it does not use). It does seem closer than the prior-guided models, but still shows an excess near the c peak and a deficit on its RHS, similar to the priors -- which is presumably a coincidence. As the priors aren't shown on the central plot, it's hard to exactly compare these, but I think a bit more discussion of the "approach" is needed to soften the conclusion, as it may be doing better but seems to be converging to something significantly far from the truth in several bins near the peak. The blue b-jet distribution is much more poorly converged and described and this should be mentioned. The point-estimate model has very similar features on the c distribution, which deserves some discussion as it's too close to be coincidence, and now the b-jet distribution is better converged (on to one of the two implied fits from the unimodal posteriors) but is not accurate other than on the position of the peak. Basically, there's a lot more going on here, with interesting grouping of models, and it's too glib just to say that the last two models have "a fair approach to the two curves" -- I think that's an overstatement of the level of improvement, particularly for the b-tag distribution.

  • Sec 3.3: the generation of the 4-jet data is not made clear. I can understand that physics process types could induce interesting correlations containing extra information, but here I think the toy model just involves sampling again from the two GN1 curves: is this correct? So what structure (other than normalisation) is there to be exploited by the statistical model that doesn't just factorise into 4 x 1D problems? This may seem obvious to the authors, but I suspect not for many readers. Is there any coherent ordering of the jets, i.e. which scores go in which tuple entries, e.g. corresponding to pT ordering of the jets? Or is that random, in which correlations would presumably be eliminated?

  • It's also not clear to me what are the relative normalisations of the two models (b and c) in either of the toy studies: depending on the signal channel of interest, and its main backgrounds in the signal region, the b and c rates could be enormously different... and this would presumably inject some physics into the mixture inference?

  • Sec 3.3: Figs 3-6 need to be brought forward to p10 onwards -- it is extremely difficult to read and understand when the text discusses figures located 6-8 pages away.

  • p10: The far-away Figs 7-10 also need to be moved closer. The text here discusses them in turn, so maybe they would work best inline, between each paragraph of discussion text.

  • Discussion: As earlier, it's unclear where the extra information is coming from in the 4-jet case. Either there are correlations in the toy-data sampling which need more explanation, or is the extra information the restriction to even numbers of b-jets? This would be a very good thing to be clear about, as it's key to understanding what information can be leveraged to make the mixture inference tractable. This is also key in the final paragraph of the discussion, which seems to be proposing a mechanism for inferring true distributions of observables -- which of course is the whole aim of differential measurements and unfolding -- but where the "multidimensionality" discussed is, I think, of a different character from that used here. (I am not convinced by this paragraph, since of course extraction of distributions this way has a lot of unconsidered prior art, but it is possible that Bayesian mixtures could add something in concert with established methods.)

  • I expected there to be a non-toy study, e.g. using MC generation for a 4t model and seeing how the methods perform on that. I suppose obtaining some approximation to a b-tag score for real MC was too difficult, but it would have been interesting. Without physics input, the paper seems to "just" be a demonstration of Bayesian mixture-inference closure on two distributions that happen to have come from b-tagging, with a little physics content in the even-Nb restriction of the mixtures in the 4-jet case. I think this is fine, but it's good to be clear about the extent to which there is physics content in this application of inference.

  • There are quite a few formatting issues in the references, e.g.:

  • [1], [2], [4], [7], [18], [23]... capitalisation issues in ATLAS, LHC, QCD, whole title, ABCD, etc.
  • [9] Different author-name formats from the rest
  • [23] T. A. Collaboration, [24] S. D. Team, ... should be proper collaboration names
  • [20], [21], [22]: missing proper experimental collaboration names
  • [21] s -> \sqrt{s} etc. and generally fix title formatting

Requested changes

See report

Recommendation

Accept in alternative Journal (see Report)

  • validity: high
  • significance: ok
  • originality: good
  • clarity: good
  • formatting: excellent
  • grammar: excellent

Author:  Ezequiel Alvarez  on 2024-09-11  [id 4765]

(in reply to Report 4 on 2024-06-16)
Category:
reply to objection

Dear Editor,

We thank the Referee for reading the manuscript and their valuable comments, which have considerably improved the manuscript.

Please find full response in attached document. New version has been submitted to Arxiv, to appear on September 13.

Ezequiel and Yuling

Attachment:

Referee_4.pdf

---

## Editorial Decision

resubmitted